# Climate Change Prevention through Community Actions and Empowerment: A Scoping Review

**DOI:** 10.3390/ijerph192214645

**Published:** 2022-11-08

**Authors:** Maria João Salvador Costa, Alexandra Leitão, Rosa Silva, Vanessa Monteiro, Pedro Melo

**Affiliations:** 1Centre for Interdisciplinary Research in Health, Institute of Health Sciences, Universidade Católica Portuguesa, 4169-005 Porto, Portugal; 2Católica Porto Business School, Research Centre in Management and Economics, Universidade Católica Portuguesa, 4169-005 Porto, Portugal; 3Health Sciences Research Unit: Nursing (UICISA: E), Nursing School of Coimbra, Portugal Centre for Evidence Based Practice: A JBI Centre of Excellence, 3004-011 Coimbra, Portugal; 4Vila Real Community Care Unit 1, 5000-557 Vila Real, Portugal

**Keywords:** community health nursing, community empowerment, community actions, stakeholders, health promotion, climate change, environmental health, local government, municipalities, sustainable cities

## Abstract

As society tries to tackle climate change around the globe, communities need to reduce its impact on human health. The purpose of this review is to identify key stakeholders involved in mitigating and adapting to climate change, as well as the type and characteristics of community empowerment actions implemented so far to address the problem. Published and unpublished studies from January 2005 to March 2022 in English and Portuguese were included in this review. The search, conducted on PubMed, CINAHL, Scopus, MEDLINE, Scopus, Web of Science, SciELO, and RCAAP (Repositório Científico de Acesso Aberto de Portugal), followed a three-step search strategy. Data extraction was performed by two independent reviewers, using an extraction tool specifically designed for the review questions. Twenty-seven studies were eligible for inclusion: six used interviews as a qualitative method, three were systematic reviews, three were case study analyses, three used surveys and questionnaires as quantitative methods, two used integrative baseline reviews, and three utilized a process model design. Six studies targeted local, public and private stakeholders. Community settings were the context target of fifteen studies, whereas twelve specifically referred to urban settings. Seven types of community actions were acknowledged across the globe, characterised as hybrid interventions and referring to the leading stakeholders: local governments, non-governmental organizations, civil society, universities, public health, and private sectors.

## 1. Introduction

Since the 1850s, the concentration of greenhouse gases (GHGs) in the atmosphere, such as carbon dioxide, methane, and nitrous oxide, has risen, mostly as a result of human activity [1]. The use of chemicals and fossil fuels in industry and inadequate land use and deforestation in agriculture have led to global warming and to the consequent climate change with the rise in both average global temperature and in sea level. Scientific evidence shows that both pose an enormous threat to human health. Examples of these are heat waves, extreme cold, flooding, dust storms, and hurricanes, among others. These major climatic issues on our planet need to be addressed, mitigated, and reduced, especially in urban areas [2].

Although the United Nations First Earth Summit took place in 1972 in Stockholm [3], Sweden, it was not until the Kyoto Protocol in Japan in 1997 that an international agreement among 160 countries was achieved. The Protocol first introduced greenhouse gas reduction targets for industrialised countries’ overall emissions of carbon dioxide and other greenhouse gases. Since then, several climate-change prevention declarations have been ratified to limit global warming by 1.5 °C. The last and more important ones were COP 21 in Paris (2015), known as the Paris Agreement, COP 25 in Madrid (2019) and COP 26 in Glasgow, Scotland in 2021, known as the United Nations Climate Change Conference [4].

According to Steffen, Richardson, Rockstrom et al. [5], three of the planetary boundaries’ frameworks presented by the Stockholm Resilience Centre (climate change, loss of biodiversity, and nitrogen use) have already been surpassed since 2009. However, by 2015 it was clear that society’s activities had pushed these boundaries, and further areas were affected, such as shifts in nutrient cycles and land use, resulting in a significant impact on human health. 

Although science shows that climate change is a major public health threat, research surprisingly demonstrates an extensive awareness of the climate concerns; however, limited capacity to adapt and change is an issue due to a lack of expertise and resources which certainly would empower communities facing climate-related vulnerabilities. 

In 2008, community intervention management models for empowerment and resilience were promoted by using key top-down interventions in which organizations outside the community, such as governments or other institutional expert agencies. However, bottom-up interventions, usually using qualitative data and involving the participation of both experts and nonexperts sensitive the conditions in each community started to be used to empower and support communities with knowledge and risk awareness [6]. Participation processes are required to involve several stakeholders in assessing community resilience and have significant benefits as they can effectively raise awareness and broaden the understanding of risks, promoting local participation at the same time [6].

Recently, a community assessment, intervention, and empowerment model (MAIEC) highlighted the importance of hybrid approaches by applying a combination of bottom-up and top-down interventions [7]. It is essential that key and effective community actions, specifically on climate change mitigation and adaptation, have integrated policies and approaches from multiple stakeholders for solutions contributing to early actions on urban governance and climate change. 

Urban areas such as cities seem to function as “human ecological systems” supported and integrated by “natural ecological systems” [8]. The interaction between these two ecological systems results in the sustainability of the urban setting as well as the health and well-being of its population. The three popular cornerstones of sustainability, first presented by Barbier [9], include environmental, social, and economic circumstances that determine the human life journey (Figure 1) as well as the physical and mental health and well-being, which is why health should be seen as a priority when planning urban sustainability policies. 

Nevertheless, to promote physical and mental health, as well as the population’s well-being, recommendations must be shared and communicated, which is why health promotion plays a significant role within community settings. Laverack acknowledges community empowerment as a product of health promotion actions, therefore strengthening the role of health practitioners. For instance, in the empowerment process, this appears to be a good strategy, as health practitioners are responsible for promoting health literacy within the community [10], key in climate mitigation and adaptation actions. 

Thus, new models for clinical decision making models such as MAIEC [7], mentioned above, may be used by health professionals such as nurses, who are environmentally aware, to plan their actions towards empowering the communities and optimizing health levels [11].

Below are some of the actions suggested to tackle climate change in urban spaces [8]:-Cross-governmental action;-Improve city planning, development, and management;-Develop integrated approaches to urban planning;-Create sustainability commissions with statutory reporting responsibilities;-Create sustainable development frameworks to guide policies;-Embed sustainability into decision making;-Ensure independent assessment of sustainability goals;-Promote health, equality of opportunities, and sustainable development.-Ensure a people-centred approach.

These actions integrate responses to health and climate change and have specific characteristic requirements for a healthy city, such as policies encouraging walking to school and local shops, easy access to public transport, provision of cycle-ways, community activities, strong public health education system and planning workforce, interdisciplinary and transdisciplinary approaches in planning, implementation and evaluation of policies, incorporation of science- and evidence-based approaches, and leadership-valuing and adaptive management system approaches. 

Governments, universities with public health and planning programs, professional organisations, industry, community organisations, business leaders, community leaders, and elected representatives seem to be some of the key stakeholders identified within the literature [8].

Thus, local governments hold key responsibilities in developing deep decarbonization plans in cities. Decarbonization is defined as the process in which fossil energy becomes just a small part of the energy mix [13] and includes targets of 80–100% net reductions in greenhouse gas emissions (GHG) by 2050 or even before.

Local governments, on the other hand, are the ones who need to take into consideration greenhouse gas emissions’ impact on the population. According to the World Resources Institute (WRI), a global research organization, which works with several governments, businesses, and other institutions to enhance people’s lives by fighting climate-related challenges, GHG emissions can be categorized in several scopes: Scope 1 emissions relate to direct emissions from sources such as on-site manufacturing and/or industrial processes, computers and data centres, and on-site transportation. Scope 2 emissions refer to indirect emissions originating from purchased electricity, steam, heating, and cooling. The source of electricity, determines whether these emissions are high or low. Scope 3 emissions refer to any indirect emissions that occur in the supply chain, such as employee commuting or business travel, purchased goods and services and use of sold products. Most of the time, Scope 3 emissions are larger than Scope 1 and 2 [14]. So, considering that urban areas, are the largest place-based source of greenhouse emissions, accounting for around 71–76% of global emissions, they are therefore a priority when it comes to community actions to mitigate or adapt to climate change [13].

Deep decarbonization in cities focuses on five significant sectors: electricity, buildings, transportation, waste, carbon sinks, and storage. In addition, innovative actions are underway to mitigate all scopes (1, 2, and 3) of greenhouse gas emissions [13].

Actions to tackle climate change can be divided into corporate actions (the ones that result from the direct action of local governments) and community actions (the ones that result from controlling GHG emissions or minimizing their impact within the boundaries of each community). All actions are focused on four priority sectors, such as energy/electricity, buildings, transportation, and waste.

As we know, climate change is expected to aggravate existing local vulnerabilities as society moves toward an increase of global trade and travel, which facilitate the arrival and dispersal of new pathogens, disease vectors, and reservoir species. Antimicrobial resistance, new viruses, and infectious diseases, animal health, food safety, and health care capacity are some of the issues local urban communities are now facing [15].

However, surveillance strategies are considered a way to provide an effective public health response. The European Parliament and Council regulates this type of surveillance through the European Centre for Disease Prevention and Control (ECDC). This agency carries on extensive literature reviews, on-going evaluation of current surveillance systems throughout Europe, risk analysis of different diseases and consultation with experts within the EU member states. This will consequently evaluate new disease risks from climate change and focus on potential changes in surveillance. This type of surveillance can be either indicator-based or event-based. Indicator-based identifies annual country-level reporting of confirmed human cases, and the event-based detects individual disease outbreaks in communities [15].

In 2012, for instance, 26 out of the required 46 infectious diseases reported by EU member states were found to be directly or indirectly related to climate change. However, being related to climate change is not in itself a cause for surveillance. Factors such as prevalence, severity, secondary complications, and human and financial costs are sustained as important parameters in a disease’s analysis, and depending on those factors, the strength of association with climate change is then categorized as low, medium, and high [15].

Actions such as enhancing collaboration between veterinary surveillance and the public health sector will ensure preparedness and more effective responses if pathogens and vectors such as zoonoses (food-borne and water-borne diseases) become a concern for humans in a specific region or community. These actions were mostly characterized by carrying out regular meetings, routine sharing of epidemiologic and laboratory data, preparation of linked response plans for human or veterinary health and coordinated outbreak investigations.

There is evidence that policy-driven adaptation actions such as an effective public health response is most likely to help contain human and financial costs derived from climate-related emerging diseases [15]. Presently, coastal areas globally are becoming unviable, with people no longer able to maintain livelihoods and settlements due to, for example, increasing floods, storm surges, coastal erosion, and sea level rise, yet there exist significant policy obstacles and practical and regulatory challenges to community-led and community-wide responses. For many, receiving support only on the individual level for relocation or other adaptive responses, individual and community harm is perpetuated through the loss of culture and identity incurred through forced assimilation policies. Often, challenges dealt to frontline communities are founded on centuries of injustices. Can these challenges to both norms and policies be addressed? Can we develop socially, culturally, environmentally, and economically just sustainable adaptation processes that support community responses, maintenance, and evolution of traditions and rejuvenate regenerative life-supporting ecosystems? These type of studies bring together indigenous community leaders, knowledge-holders, and allied collaborators from Louisiana, Hawaii, Alaska, Borikén/Puerto Rico and the Marshall Islands to share their stories and lived experiences of the relocation and other adaptive challenges in their homelands and territories, the obstacles posed by state or regional governments in community adaptation efforts, ideas for transforming the research paradigm from expecting communities to answer scientific questions to having scientists address community priorities and the healing processes that communities are employing. The contributors are connected through the Rising Voices Centre for Indigenous and Earth Sciences, which brings together indigenous, tribal, and community leaders; atmospheric, social, biological, and ecological scientists; students; educators and other experts and facilitates intercultural, relation-based approaches for understanding and adapting to extreme weather and climate events, climate variability, and climate change [16].

In coastal areas, for instance, Maldonado and others highlight that there is a singularity to each community; therefore, prior to any plan for relocation, site expansion, climate change adaptation or mitigation, we must ensure policies and processes are humanized, guaranteeing that communities’ unique histories, traditions, and priorities are properly acknowledged [16].

When climate change migration or displacement occurs, communities require support to continue their practices and traditions elsewhere as this will certainly contribute to their resilience and prevent a loss of identity or any forced assimilation. Community resettlement entails the need to listen not only to scientists, agencies, and policy makers but also to citizens, enabling them to contribute to all decisions [16]. 

Despite existing evidence showing that local government climate policies have minimal impact and do not reduce GHG emissions, due to financial constraints, there is still a lack of research regarding climate policy-making processes and the role of key stakeholders. Our study is designed to fill this gap. Acting alone, local governments will be unable to mitigate climate problems. Instead, local activists, such as climate protection networks and other grassroots groups, may push elected representatives to act, build partnerships, and gain required public support. New research is now showing that bottom-up policy processes such as the ones above can develop new climate policies based on new standards and programs, which have more impact on reducing GHG emissions [17].

The European Commission has recently adopted the concept of “nature-based solutions” (NbS) that aims to be inclusive, transparent, and empower governance processes. NbS has shown its benefits when it comes to dealing with the challenges related to economic viability, environmental protection, and social equity. An example of this is the Horizon 2020 funding programme, responsible for sponsoring several research programs and teams focused on the verification, design, and development of NbS to encourage its implementation [18].

A wide range of stakeholders at different governance levels need to be involved to generate collective community actions which then lead to more sustainable approaches [18]. NbS activities promote “social cohesion, citizen security, environmental justice, and human health” [18] (p. 2).

To build resilient cities that can meet the challenges of natural hazard management, for instance, Thaler et al. [19] suggest that natural hazard dynamics need changing to ensure better urban environments and promote community well-being. Key individuals and groups of activists are often engaged as policy entrepreneurs, motivating the community to engage in societal transformation that goes beyond consultation and information-sharing. 

The example of Australia regarding adaptation strategies is key. Rychetnik, Sainsbury, and Stewart [20] refer to the need of preparedness of local health districts in responding to the inevitable effects of climate change. Available data and models of climate change impact assessments must be used to identify existing risks and vulnerabilities. This will help health services prepare and respond to health emergencies and disasters in the future. Only by doing so, can longer-term financial costs be avoided and a safer environment and better health care be provided.

Informing what stakeholders are involved in preventing and adapting to climate change and how they are accomplishing this colossal task is key. Therefore, prior to writing our scoping review protocol [21], preliminary search was conducted on PROSPERO, MEDLINE, the Cochrane Database of Systematic Reviews and the JBI Evidence Synthesis. This search did not find any review that met the current objective. Hence, the present scoping review explored the variety of stakeholders participating in community empowerment processes to tackle climate change and their community actions towards it. The objective was to identify key stakeholders involved in mitigating and adapting to climate change and the type of actions led or implemented so far, including the characteristics of these, whilst focusing on urban community settings. As per prior protocol, studies were limited to the period between 2005 and March 2022, although the authors have assumed that recent studies seemed more relevant for current practices. 

## 2. Review Questions

The scoping review focused on the questions below:Which community empowerment actions have been implemented so far to prevent climate change?What are the characteristics of these community actions to prevent climate change using both adaptation and mitigation approaches?Which stakeholders led or implemented these community actions?

## 3. Inclusion Criteria

### 3.1. Participants

This review considered studies involving stakeholders such as leaders, organisations, governments, managers, health professionals, and others who led or implemented community actions in preventing climate change through mitigation or adaptation.

### 3.2. Concepts

This review explored the combination of two core concepts: community empowerment and climate change. Although community empowerment is a very used term, its meaning may change according to the context and depending on the stakeholders involved, but Laverack [10] highlights that it improves participation, develops local leadership, enhances the capability to query the state of things and the capability of organizing resources for an improved management of people and organizations. For the present review, we used the definition of climate change used by United Nations [22] which attributes climate change directly or indirectly to human activity, with consequences for the global atmosphere, inducing climate variability over time.

### 3.3. Contexts

Considering the project in which this investigation is being carried out, this review considered all studies that include community actions led or implemented by stakeholders in any urban community environment. All urban community settings, such as houses, institutions, cities, and others, were considered if they included community actions that led to climate change mitigation or adaptation.

### 3.4. Type of Studies

Quantitative, qualitative, and mixed methods studies were included for consideration, as well as texts, opinion papers, and other grey literature such as dissertations, published or unpublished. Relevant documents retrieved by reference screening were also considered.

## 4. Materials and Methods

This review followed the JBI methodology for scoping reviews and was carried out in accordance with the Preferred Reporting Items for Systematic Reviews and Meta-Analyses extension for Scoping Reviews (PRISMA-ScR) checklist [23]. (Please refer to Appendix A). As previously mentioned, an *a priori* protocol was conducted and published earlier in 2022 where objectives, inclusion criteria and methods of analysis were identified and detailed in advance.

### 4.1. Search Strategy

The authors used a three-step approach for the search strategy:A preliminary search was initially conducted to develop the prior protocol on the following databases: Prospero, MEDLINE, the Cochrane Database of Systematic Reviews and the JBI Evidence Synthesis, aiming to identify relevant articles on the topic.A comprehensive search was then followed using search terms, keywords, and MeSH descriptors, where titles and abstracts of eligible articles on the topic were considered. (Please refer to Appendix B).The third step consisted of the screening of reference lists from the articles to add further relevant studies that would have been missed otherwise (snowballing sampling) as this technique represents a “purposive method of data collection in qualitative research” [24] (p. 1). Finally, only free, full-text articles, written in English or Portuguese, published from 1 January 2005, until 7 March 2022 were considered for inclusion.

#### Information Sources

The abovementioned databases (PUBMED, MEDLINE, CINAHL, Scopus, Web of Science, SciELO) were used to locate both published and unpublished studies. Unpublished studies and grey literature were also included through RCAAP (Repositório Científico de Acesso Aberto de Portugal), and hand-searched references were considered for inclusion. Google Scholar was not used since it does not retrieve all evidence-based studies, and the platform only allows a limited number of documents. 

### 4.2. Study Selection 

Although articles published since 2005—the year in which the Kyoto Protocol (1997) came into force—were considered for inclusion, a lot has happened since, which is why the authors worked to select the most relevant and up-to date studies, as they reflect the most current practices.

Following the search, all relevant studies identified from 2005 onwards were identified against the inclusion criteria and uploaded onto Rayyan [25], a web-tool for a thorough analysis of the selected studies, to identify duplicate records and resolve each one using a blind process with the objective of retrieving the relevant studies for the current review. A first double-blind review of titles and abstracts was conducted in detail by two independent reviewers against the inclusion criteria. Finally, the free and full-text relevant studies eligible were comprehensively reviewed, and relevant ones were selected for inclusion. All disagreements were resolved through discussion, involving a third reviewer when required. The search retrieved a total of 2543 records. Of these, 438 records were duplicate studies and therefore deleted, resulting in 2105 eligible to be screened alongside 2 others from snowballing. From the 39 free full-text articles read, 12 were excluded (9 due to ineligible outcome and 3 due to ineligible context). In total, 27 studies were retrieved to be included within the narrative synthesis. (Please refer to Figure 2).

### 4.3. Data Extraction

Data were extracted from the studies included in the scoping review by teams of two independent reviewers per year of studies. A data extraction tool, already developed in the review protocol, now enhanced, was used by each team. The data extracted included detailed information about stakeholders involved and the type and characteristics of the actions identified in promoting community empowerment aiming to prevent climate change in our planet. Disagreements were resolved through dialogue and discussion amongst each team of reviewers.

A data extraction tool aligned with the objectives and the aim of the review questions containing the abovementioned key aspects of the studies selected for inclusion is provided below to support this review. (Please refer to Appendix C).

### 4.4. Data Analysis and Presentation

As per recommendation from the JBI scoping review guidelines, data collected with the data extraction tool is presented graphically, however, complemented by a narrative summary of the results describing how these relate to the review’s objective and questions. The data generated was analysed through frequency and text analysis to identify the categories and characteristics of community actions that led to or were implemented as well as the types of stakeholders involved in each one of them, aiming to respond to the review’s objective and questions.

## 5. Results

### 5.1. Study Inclusion

In total, the database search and other sources retrieved 2543 studies, 438 duplicates were removed, and 2105 were left to assess eligibility with another two additional papers considered for inclusion using snowballing. Two independent reviewers screened all titles and abstracts, and 39 free full-text papers were selected for a comprehensive reading, from which 12 were excluded due to ineligible outcome or context. (Please refer to Appendix D).

The 27 studies selected for inclusion were exported from Rayyan to a free online version of Zotero, a references management tool far more effective than Endnote or Mendeley, as recently their last versions have been causing some disruptions. 

The results of the search are fully reported in the present scoping review and presented in a Preferred Reporting Items for systematic Reviews and Meta-Analysis (PRISMA) checklist [23], which will clarify the process. (Please refer to Appendix A).

### 5.2. Characteristics of Included Studies

A total of 11 of the 27 studies were from European countries, 3 of which were conducted in the United Kingdom (UK), 1 of them in partnership with the Netherlands, and 1 with Belgium. However, most studies were from the American continent, seven of which were from the United States (one of them in partnership with Brazil), five were from Canada (one of them in partnership with the Netherlands and the UK), two were from Australia, and one study was from the Philippines. There is also one study from South Africa. (Please refer to Figure 1).

Regarding the year of the studies, and although the inclusion criteria focus on studies since 2005, the authors paid special attention to more contemporary data and found 20 relevant studies from 2015 onwards, two being from 2022, four from 2021, two from 2020 and five from 2019. The minority of studies, although relevant, refer to 2008 up to 2012. 

Only 6 of these 27 studies used interviews as a qualitative method [13,17,19,26,27,28]. Three studies were systematic reviews [29,30,31], three were case-study analyses [18,32,33], another three used surveys and questionnaires as quantitative methods [34,35,36], two used integrative baseline reviews [2,37] and another three studies utilized a process model design as a method [20,38,39]. In eight studies [15,20,26,28,30,31,35,40], the main aim was to assess public health adaptive approaches and preparedness to tackle climate change; in five studies [17,18,19,27,34], the main aim was to discuss current city policies and solutions on the topic.

**Figure 2 ijerph-19-14645-f002:**
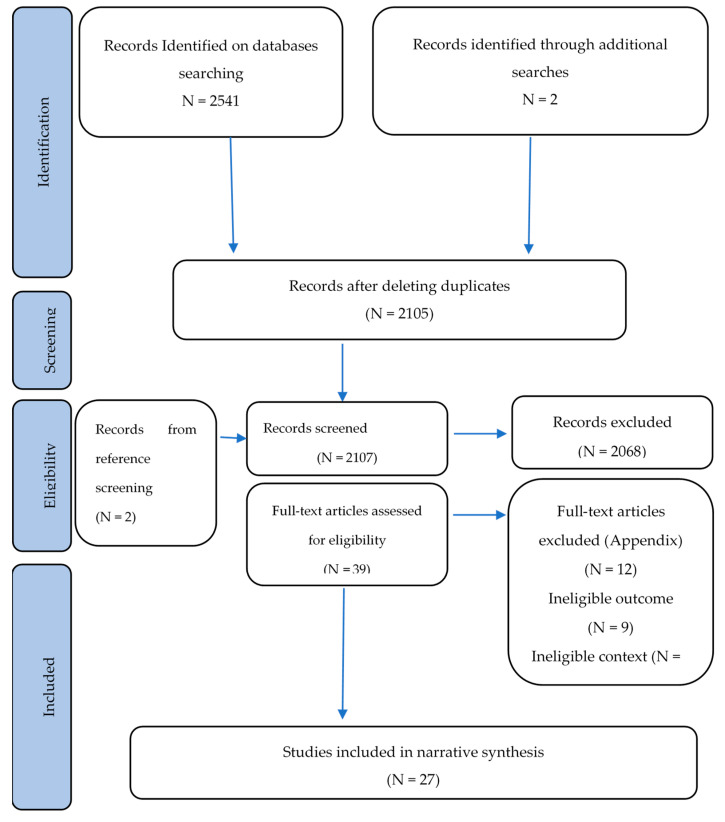
Flow diagram for the scoping review process adapted from the PRISMA statement by Moher and colleagues [41].

### 5.3. Review Findings

#### 5.3.1. Populations

The authors identified seven groups of populations in the present review. Six studies refer to one of them: local, public, and private stakeholders [18,19,35,39,42,43]; four of them focus on the population group of municipalities or local governments [13,28,31,42]; four target the population group of cities and city council representatives [27,29,38,44]; another four focus on public health services and professionals [20,26,36,40]; scientists and experts were also part of the focus of one particular study [27], and finally citizens [34] and community leaders [16] were also the population of choice in two other studies. It can be said that the studies included have mainly targeted local, public, and private stakeholders as their population [18,19,35,39,42,43], followed by public health services [20,26,36,40], local and regional governments [13,28,31,38], city authorities [27,29,38,44], scientists and experts [27], and citizens [34] or community leaders [16].

#### 5.3.2. Contexts

The selected studies had “community settings” as an inclusion criteria requirement, hence we can confirm that 55,6% of the studies selected for inclusion focused on community contexts [15,16,17,19,20,26,28,30,32,35,36,37,40,42,43], and the remainder of them were specifically targeted to urban settings [2,8,13,18,27,29,31,33,34,38,39,44]. Rural contexts were excluded due to inclusion criteria. All findings were categorised according to the review questions.

#### 5.3.3. Review Question 1—Types of Community Actions

For Review Question 1, detailed key information about each community action led or implemented by stakeholders differ among studies, mostly depending on the type of stakeholders involved. Below, and following Bardin’s [45] concept of “*categories a posteriori*”, readers may find a synthesis obtained once results were interpreted, with a presentation of inferences in the form of final categories for community actions contributing to the empowerment of communities, found in the present review. (Please refer to Figure 2):**Actions of political scope** [8,13,16,17,19,20,26,27,28,29,31,33,34,37,40,43,44] (local, regional, national, and European) are referred to in 17 of the selected studies; mainly, these actions include top-down interventions such as city management and planning, local/regional/national/European policies and control, recommendations, energy solutions, urban planning (street widths, off-street trails for cyclists and pedestrians, neighbourhood playgrounds and parks, sporting and recreational facilities), implementation of policies that support physical activity and active living, improved laws and traffic safety, adaptative measures to extreme weather and other climate variability, projects to protect residents, integration of nature-based solutions that improve microclimate, limit urban heat and improve air quality, increasing green urban spaces, promote active transport, fiscal and regulatory measures, underground parking, leisure parks, flood defences, review flood maps, review cooling capacity on local facilities and backup generators, review insurance policies, claiming leadership, implementing low carbon society, work in partnership with multistakeholder groups, translate science to lay audience, evaluate local adaptation plans, improve public transport networks, mandatory vehicle inspections, road charges, reduce speed of vehicles, and walking to school policies, among others.**Actions of community scope** [2,8,16,17,19,26,28,31,32,33,39,40,42,43] mentioned in 14 of the selected studies mainly include bottom-up actions such as the ones resulting of social movements and cohesion by maintaining the community identity and culture, as well as actions resulting from partnerships, capacity building, and interagency efforts at the community level. Actions such as democratic participation and risk awareness programs are also a key part of the community scope.**Actions based on public health and environmental health** [2,8,16,18,19,27,30,31,34,35,36,37,38,40,44] are also referred to in 15 studies, mainly as hybrid actions, a result of the combination between top-down and bottom-up interventions; these actions involve public health and environmental health promotion and implementation of programs, health sector leadership related actions as well as analysing sub-district vulnerabilities, emergency preparedness, warning and observation systems, monitoring processes and risk awareness, promoting cycling and walking, encouraging children to play, marketing through media the benefits of physical activity, promoting new workplace practices to prevent sedentarism at work, promoting exercise within vulnerable groups, providing education, and training for intersectorial sustainability, improving epidemiological surveillance, assessing heat vulnerability in homebound populations, ensuring mosquito surveillance, implementing interventions in faith-based communities, monitoring severe weather predictions, educating population on health and safety on extreme weather events (through websites, webpages, hotlines, and others), encouraging health staff to collaborate with universities on climate change adaptation, coordinating for shared knowledge, raising awareness to climate change, providing updated international guidelines on heat-health plans, and ensuring surveillance of endemic and emerging diseases, as strategies to the empowerment of the community on climate-related mitigation and adaptation.**Actions based on resource management** [8,13,17,20,37,40,44] are mentioned in seven of the studies and mainly refer to housing energy efficiency measures, an equitable allocation of energy resources, known as energy democracy, that leads to sustainable consumption, reflecting the abovementioned combination of top-down and bottom-up (hybrid) actions. The use of renewable energy and water conservation systems are key in managing resources effectively.**Actions based on science and research** [16,18,27,28,31,33] are cited in six studies referring to an investment in informing communities and communicating climate action science and research results that can support populations with mitigating climate change impacts or promoting adaptative strategies. Research actions promote change, involving and empowering communities in developing new behaviors and climate action practices, and providing technical support for multisector planning efforts, which is why such examples are mainly presented as hybrid approaches.**Economy-based actions** [19,39,42] have references in three studies, mainly as a result of a combination of both top-down and bottom-up actions as they relate to adaptation strategies and improvement in production processes, sustainable business practices, sustainably produced food, use of sustainable technology and fuel improvements, introduction of electric cars, increase in cash income, among others.**Funding-related actions** [16,19,33] refer to fundraising, affecting all levels (local, regional, national, and European) and attracting financial investors and developers from all areas to engage with the research sector to adapt strategies and with society in general, sharing key information for the empowerment of communities. These are referred to in three studies and mainly represent a combination of both top-down and bottom-up actions.

#### 5.3.4. Review Question 2—Characterisation of Community Actions

For Review Question 2, the authors concluded that all types of community actions mentioned above can be divided according to their characteristics. Consequently, and agreeing with the models’ discussion by Melo [7], political actions (of local, regional, national, or European scope) are mainly characterised as top-down interventions. Community actions mainly reflect the bottom-up model of interventions; and public health and environmental actions, science and research actions, resource management actions, economy-related actions, and funding-based actions, all mainly point to a hybrid approach of community interventions. In the present review, 85% of the studies display a total of 42 references to top-down approach interventions, 63% percent display a total of 19 references to a hybrid approach, whilst 18,5% display a total of 5 references to bottom-up actions. (Please refer to Figure 3 below).

#### 5.3.5. Review Question 3—Stakeholders

For Review Question 3, 14 categories of stakeholders were identified, 4 of which were more cited than others as they are more frequently involved in climate change prevention by leading or implementing a diverse range of community actions and community empowerment.

Therefore, most of the studies selected, 21, refer to local governments and municipalities as key stakeholders [8,13,17,18,19,26,27,29,30,31,32,33,34,35,37,38,39,40,42,43,44]. These are then followed by the civil society [8,16,17,18,19,32,33,34,35,38,39,42,43], namely community associations and leaders, non-governmental organizations (NGOs) and volunteers, and citizens being mentioned in 13 of the selected studies. Universities and academic settings [8,16,18,27,32,35,40,42,44,46] follow, as they play a significant role as well, being mentioned in 10 of the selected studies. Equally referred to in nine of the selected studies, the public health sector [2,20,26,28,30,35,36,37,38], and the private sector [8,13,18,33,34,38,39,40,43] both stand out as key elements in climate change adaptive capacity or climate change mitigation strategies. Within the public sector, the studies referred to heads of departments, public health professionals, and officials. The private sector references are particularly made of companies in general, banking, other financial institutions and business leaders and developers. The other nine categories are listed as follows: national/federal government [8,20,26,28,34,40], regional/provincial/state government [8,28,40], professional organisations and unions [8,19], school representatives and educators [16,34], emergency services and firefighters [38], European Parliament/Commission/Council [15], European Centre for Disease Control and Prevention (ECDC) [15], transport sector [37], and electrical and fuel providers [2]. (Please refer to Figure 4 below).

## 6. Discussion

The comprehensive literature search of articles published between 2005 and 2022 retrieved 27 relevant studies identifying several categories of community actions, as well as stakeholders involved aiming to mitigate or adapt to climate change. Although the number of studies encountered seems high, none of them presents an overview of the global climate attitude within communities, hence the reason for the present review.

Our study reinforces existing but limited literature on a global perspective for community plans and actions to mitigate or adapt to climate change- (please refer to Figure 3). Strategies such as deep decarbonization plans and actions are currently underway around the globe, and the present review demonstrates how and by whom these can be developed and implemented at local, regional, national, and international levels, whilst overcoming local authority limitations through leveraging partnerships, allowing mitigation targets to be reached.

Therefore, Gimenez et al. [38] discuss the importance of assigning climate-related coordination responsibilities to a specific department, by setting up a multi-stakeholder group, involving local government, emergency services, firefighters, officers, public health managers, researchers, citizens, and others who can work together, applying research-based actions for multisector collaborative plans in tackling climate change [32].

Although in some locations this has not yet been implemented in a structured way, generally, the main objective of stakeholders is to build resilient and climate-neutral cities [33]. However, where higher-level governments are not likely to tackle climate change, local elected representatives and citizen groups tend to adopt impressive climate strategies [17].

Unfortunately, in 30% of the cases, mitigation and adaptation processes are linked to the political will [26] at several levels, which means that sustainable and resilient solutions may not be implemented.

Nevertheless, local authorities with a leading attitude seem to focus on five priority sectors: electricity, buildings, transport, waste and carbon sinks, and storage [13] to reach their GHG mitigation targets. Therefore, sustainable transport policies [39], restrictions on the use of cars, improvement of the public transport networks, tolls [29], electric cars [44], improved production systems [42], pedestrian friendly policies [42], and other environmental policies [27] are some of the political actions in place.

The modern way of life, though, entails elevated levels of inactivity and the absence of outdoor urban environment policy; both lead to a wide range of chronic diseases. When healthier and environmentally friendly urban spaces are made available for leisure time for physical activities, these meet the citizens needs and desires for green active living [34] and foster local social cohesion, avoiding polarization and health disparities. This is confirmed by Arlati et al. [18] when referring to NbS as of great interest among community actions, contributing to unite people within their neighbourhoods and cities, promoting resilience and mutual sustainable growth. Thaler et al. [19] corroborate these types of actions referring to the implementation of NbS as key political actions and to urbanization planning actions, such as underground parking, as fundamental to release the surface for communal activities such as parks, cycling roads, etc.

However, the world needs to remember that each community has its unique background, culture, traditions, and priorities, and when it comes to adaptation processes, local authorities and other stakeholders need to acknowledge these variants as key. As an example, people from the island nation of Tokelau [16], hit by a hurricane in the 1960s, were forced to relocate, resulting in a loss of identity, language, practices, culture, and way of life. Nonmaterial elements such as the ones mentioned above are fundamental within communities and require acknowledgment prior to climate-related actions, such as relocation plans. In these cases, research-based actions need to guide political actions supported by sectorial experts, local development agencies, academia, and civil society.

Therefore, it is unmistakably noted that the ongoing development and implementation of community actions towards climate change are still mainly characterised by singular top-down imposition; nevertheless, a growing trend is now building, reflecting a higher focus on hybrid approaches [38], where the active participation of key community-based stakeholders, other that public institutions, seems to follow a combination of both top-down and bottom-up models of intervention in the community.

Although we seem far from the ideal reality, a wide range of initiatives are now aligned with what is known as “*triple-duty actions*, and local level engagement seems essential to fight climate change [37]. The Organization for Economic Co-operation and Development (OECD) claims that policies in areas such as transport, food, and energy should be focused on the community’s ecological and human well-being by using the triple win climate lens framework [46], as per Figure 4 below.

The governments of Scotland and New Zealand are also applying “economies of well-being”, and the Lancet publication [47] on food and consumption advocated for more “triple-duty” actions. This is aligned with Raworth and his “doughnut economy” [48], also consistent with the concepts of planetary boundaries [5].

When it comes to the well-being of people, several authors refer to public institutions, such as local health departments, as relevant stakeholders. By using a top-down intervention model, they are in the best position to assess the community’s vulnerability by assessing homebound populations and their vulnerability to extreme heat, by improving epidemiologic surveillance, by developing mosquito surveillance [35], by monitoring infectious diseases [15] among others. Likewise, other studies highlight the importance of the health sector as a climate-related stakeholder as it can provide appropriate training for healthcare workers and public health practitioners, developing expertise and raising awareness on disaster preparedness and response, key strategy plans to face the climate challenge [30]. Rychetnik et al. [20] emphasised that using opportunities to enhance health information on extreme weather events or by leading direct public education on climate-related risks and health and safety seem to be a part of the key role of public health services as climate stakeholders, who can use data sources and models to predict, prepare for, and respond to those risks. Similarly, these authors highlight the importance of building collaborative relationships with other stakeholders, such as academics, bringing together health staff and university experts and researchers working to develop CC adaptation plans. Other authors corroborate this by highlighting the need for disseminating usable knowledge [28], raising climate-related awareness [43], contributing to effective communication and intervention strategies to reduce heat-related mortality [2], influencing community practices and behaviors [31], and informing, educating and empowering people about climate-related health issues [40].

However, the public health sector can extend actions even further by using media marketing to recommend reducing energy use and greenhouse gas emissions [36], working alongside the resource management sector to ensure water conservation and with local business stakeholders to promote sustainable business practices in the community [39].

## 7. Limitations of the Study

Although the present study could have been enhanced with the inclusion of an international database and could have included further languages other than Portuguese and English, the authors believe that the objectives of the present work were achieved as they map the state of the art and existing literature on the topic; therefore, despite its limitations, a scoping review was the appropriate method to achieve the conclusions below.

## 8. Conclusions

The present review identified 27 relevant studies responding to the objective and to the review questions initially stated in our work. Data analysed and extracted show that further research is required to identify the effectiveness of climate-related stakeholder collaboration processes, specifically in urban contexts, and to develop a shared and global strategic guidance tool in tackling climate change within communities.

Seven types of community actions were acknowledged as being led and implemented across the globe, including political scope, community scope, public health and environmental health, resource management, science, and research, economy-based, and funding-related. Political actions such as the ones developed by local government authorities, community actions such as the ones led by NGOs and other organisations and the actions of public health and environmental health are the leading stakeholders, alongside universities and the private sector.

The community intervention models used were mostly focused on a hybrid approach by using a combination of both top-down and bottom-up intervention models. Public health services and professionals, alongside scientists and experts, city council representatives, citizens, and community leaders play vital roles.

Public health authorities are key stakeholders in increasing populations’ resilience to the health impacts of climate change and in contributing to reduce the population’s vulnerability. As local public authorities, they hold full knowledge of the local population and full awareness of the localized nature of climate change impacts.

We can conclude that comprehensive public participation should always be facilitated and fully recommended as the only way to inspire communities towards making a difference, acting, and engaging in decision-making processes for a sustainable development by 2030 and carbon neutrality by 2050.

It is therefore vital that future government authorities continue to inform, consult, and engage public and private stakeholders through meetings and public conferences to bring together shared decision processes. It is also vital to continue to monitor the participatory process and ensure transparency and active participation from all parties involved. This will allow the identification of gaps and necessary adjustments. Finally, a full assessment of the whole participatory process, ensuring the engagement of all stakeholders must be conducted.

Involving experts to guide how to maximize environmental benefits from a public health perspective, implementing specific actions within the community and using intersectoral design teams will certainly empower communities in improving energy efficiency but mostly in achieving sustainable living targets with a positive impact on people’s health.

## Data Availability

For data supporting reported results please contact the authors of this review.

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
