# Peer review of "Climate Change Prevention through Community Actions and Empowerment: A Scoping Review"

_ijerph, 2022, doi:10.3390/ijerph192214645_

Round 1

Reviewer 1 Report

The potential for the proposed research could be significant, however, the research must actually be completed for a meaningful assessment. There are some significant concerns as follows:

  1. A scoping review protocol for this type of research is inappropriate and unnecessary since there are established methodologies for such reviews.
  2. Undertaking a review paper will require ongoing refinement and multiple changes due to the iterative nature of the process, especially as it pertains to scoping reviews therefore this "protocol" should be integrated into the "methods" section of the proposed review paper.
  3. The premise of this review is fundamentally flawed as:
    1. Stakeholder involvement in Health Promotion and Environmental Health policy, initiatives, and activities IS standard practice across multiple jurisdictions.
    2. The scope of the proposed review is too broad and should be confined or at least organized geographically and regionally. Additionally, the authors should define and refine the types of actions for inclusion: for example, are they focusing on public health actions, environmental management actions, sustainable development actions, etc.?
    3. The link between climate change and its impact on human health both directly and indirectly has not been established in the proposed review.  There is a lot of literature on this topic and the authors should make use of it as it will help focus the scope of the review.
    4. The linkages between community health nurses and climate change action is tenuous at best - evidence should be provided for such a statement.
    5. The value and novelty of this review needs to be established - what does it add to the existing discourse? There are many studies that support community empowerment for climate change mitigation and adaptation. I respectfully suggest the authors review the following papers for examples: https://doi.org/10.3390/su13010379 and https://doi.org/10.3390/ijerph18115763.

Author reply:

Response to Reviewer 1 Comments

Dear Reviewer,
We would like to thank you for your constructive feedback and comments. The authors will respond below, point by point, based on the comments made.
Point 1: A scoping review protocol for this type of research is inappropriate and unnecessary since there are established methodologies for such reviews.
Response 1: Firstly, we would like to mention that one of the authors of this paper is a member of the Joanna Briggs Institute (JBI) core staff. JBI is an international research organization that develops review methodology, delivers unique evidence-based information, software, education, and training designed to improve healthcare practice and health outcomes1.
For this purpose, JBI has a manual https://jbi-global-wiki.refined.site/space/MANUALwith different types of methodologies to conduct systematic reviews.
Within the different methodologies that we know (possibly more related to the disciplinary areas of health, such as Systematic reviews of qualitative evidence or Umbrella reviews/ Overviews of Reviews) we have identified that a scoping review is the most suitable for the questions we intend to answer in this review.
The authors have chosen a scoping review as it has been defined by JBI Manual for Evidence Synthesis (2020)2 as a means to “map the key concepts that underpin a field of research, as well as to clarify working definitions, and/or the conceptual boundaries of a topic (Arksey & O’Malley 2005). A scoping review may address one of these aims or all of them. A scoping review of scoping reviews found that the three most common reasons for conducting a scoping review were to explore the breadth or extent of the literature, map and summarize the evidence, and inform future research (Tricco et al. 2016b). The indications for scoping reviews are listed below: (Munn et al. 2018a)
• As a precursor to a systematic review.
• To identify the types of available evidence in a given field.
• To identify and analyse knowledge gaps.
• To clarify key concepts/ definitions in the literature.
• To examine how research is conducted on a certain topic or field.
• To identify key characteristics or factors related to a concept.”1
1 https://jbi.global/
2 Peters MDJ, Godfrey C, McInerney P, Munn Z, Tricco AC, Khalil, H. Chapter 11: Scoping Reviews (2020 version). In: Aromataris E, Munn Z (Editors). JBI Manual for Evidence Synthesis, JBI, 2020. Available from https://synthesismanual.jbi.global. https://doi.org/10.46658/JBIMES-20-12

The aforementioned is the reason why the authors considered the present review could help mapping the type/characteristics of community actions led or implemented so far in tackling climate change as well as the key stakeholders involved.
As the authors intend to " map community actions led or implemented so far, aimed for the community empowerment in preventing climate change" we understand that this methodology responds to the requested, however we are open to any other suggestion that the reviewer wishes to add.
Point 2: Undertaking a review paper will require ongoing refinement and multiple changes due to the iterative nature of the process, especially as it pertains to scoping reviews therefore this "protocol" should be integrated into the "methods" section of the proposed review paper.
Response 2: Thank you for the suggestion. Indeed, knowledge has a perennial nature, but mapping community empowerment actions implemented within the context of climate change is a necessary knowledge to guide practice in the community. The protocol, as well as the method recommended by JBI Manual for Evidence Synthesis, will be strictly followed. Any deviation from the protocol will be clearly explained in the scoping review report and in future publications that may follow.

Point 3: The premise of this review is fundamentally flawed as:
a. Stakeholder involvement in Health Promotion and Environmental Health policy, initiatives, and activities IS standard practice across multiple jurisdictions.
b. The scope of the proposed review is too broad and should be confined or at least organized geographically and regionally. Additionally, the authors should define and refine the types of actions for inclusion: for example, are they focusing on public health actions, environmental management actions, sustainable development actions, etc.?
c. The link between climate change and its impact on human health both directly and indirectly has not been established in the proposed review. There is a lot of literature on this topic and the authors should make use of it as it will help focus the scope of the review.
d. The linkages between community health nurses and climate change action is tenuous at best - evidence should be provided for such a statement.
e. The value and novelty of this review needs to be established - what does it add to the existing discourse? There are many studies that support community empowerment for climate change mitigation and adaptation. I respectfully suggest the authors review the following papers for examples: https://doi.org/10.3390/su13010379 and https://doi.org/10.3390/ijerph18115763.

Response 3 a: The authors have acknowledged your points and agree that the involvement of stakeholders seem to be common practice across the globe, however, one of the review questions aims precisely to identify which stakeholders are involved in tackling the challenge of climate change (either by mitigation or adaptation strategies) which we believe explains the present review. This will help mapping participants into the next steps of our larger study.

We take this opportunity to inform you that this review is part of a larger project, funded and based at the Universidade Católica Portuguesa, which involves different research centres (arts, economy, psychology and education and health sciences), already aproved and financed by the European Regional Development Fund (FEDER) and the Foundation for Science and Technology (FCT). The project includes three lines of research: one line, based on Heritage Protection, the second line, more connected with art and education and the third one, where this study is included, related to community empowerment and social economy.

Response 3 b: The authors appreciate your comment, although they wish to kindly remind that Scoping Reviews intend exactly to achieve broader perspectives. Tthe present review aims precisely to identify which are the characteristics of these community actions implemented so far in tackling climate change (Hopefully an outcome of this review by identifying/dividing it by types) globally.

We are currently carrying out a global mapping but will use the results identified in this first study to leverage in future a locoregional study involving three large municipalities from the northern region of Portugal, consistent with our project line of research, related to community empowerment and the social economy.
Response 3 c: Thank you for your comment. Please refer now to new lines 81-84, where the authors intend to briefly address your comment acknowledging the impact of climate change on human health more clearly (although having in mind the word limit at stake), specifically by stating that “These last years, the hottest in history, also resulted in a disease- friendly environment worldwide. For instance, Vector-borne diseases such as Zika are now starting to spread in cooler northern regions and Covid 19, the current pandemic, has also taken advantage of a planet where climate change has not been properly acknowledged [8].”, which we trust adds contemporaneity to the article.
Although the impact of climate change on human health is not the aim of the present review, it remains central, and the authors agree to the importance of the above statement and wish to further expand the subject on the actual Scoping Review, as this is still its Protocol.

Another polite reminder that this protocol, is part of a larger project, where we will develop further studies, more focused on health care, namely in the context of Community Health and Public Health

Response 3 d: Thank you for your comments. Perhaps it´s still not clear to the reader the key role of Community Health Nursing in tackling Climate Change, as it is limited to a single paragraph due to the protocol word limit. However, we have now rephrased it (Please refer to new lines 110-115) Community empowerment is key in Health Promotion actions [18] and health practitioners within the community empower people on health literacy [19]. Nurses, for instance, are key in community environment awareness and may plan their actions according to a new model for clinical decision. The Community Assessment, Interven-tion and Empowerment Model (MAIEC) assesses and enables actions to empower in-dividuals and optimise their health [20]. – page 3 of the document.

The authors are planning to expand this further throughout the scoping review and explore the breadth and extent of the literature, mapping and identifying community actions (types and characteristics) but also stakeholders involved, informing future research of knowledge gaps in the field. According to Pedro Melo (2020) a new model for clinical decision -MAIEC- assesses and enables actions to empower communities in optimizing their health. Undoubtedly, optimizing community health and promoting community environment awareness is key within Community Health Nurses´ role.
MAIEC's clinical decision matrix enables the possibility to diagnose, in the context of community management, in this case associated with climate change, the community process, community participation and community leadership. Considering Community Health Nursing care and the existence of specialist nurses in this field, there is a potential for them to be identified as key partners in other scientific and professional areas around environmental issues.
Response 3 e: Thank you for your kind suggestions. We have acknowledged these two papers which seem to be key in the field as well as contemporaneous and we all trust that citing it within our scoping review will certainly increase its pertinence. We thank you for putting us in contact with these researchers and fields of study, who we believe might even become our international partners!

The novelty of the results we intend to reach, after mapping community actions and stakeholders, is precisely to identify the contributions that specialist nurses in Community Health Nursing can bring, considering the competencies we have previously described, in order to contribute to community empowerment and the inherent community actions, with appropriate stakeholders (either by mitigation or adaptation strategies). This will wisely allow an adequate use of health resources, namely specialist nurses, to work in partnership with other leaders and specialists in the area, considering the central importance of environmental issues, as we understand that the esteemed reviewer is so framed and sensitized.

Conclusions:
The authors feel that you have added value to our manuscript and we thank you for your valid points, now addressed.
We have requested a revision from an English native lecturer to comply with your request, ensuring excellent language standards, despite the fact the first author of the present protocol has lived and worked in London, UK since 2014. Maria João Costa is a Fellow of the Higher Education Academy, currently HE, following completion of a PGCE at Anglia Ruskin University in Cambridge in 2018.
Nevertheless, all suggestions were considered and the revised manuscript shows all changes.

Reviewer 2 Report

Climate Change Prevention through Community Actions and Empowerment: A Scoping Review Protocol

This manuscript addresses important issues and does a more than adequate job of providing potential readers with history of the topic, the importance of the topic, and the need for multiple disciplines to contribute to addressing the topic. Readers are provided with information about the various climate change declarations and their implications for research that is needed if the identified problems are to be addressed. The need for enhanced partnerships and support among countries and what this suggests for research is also articulated. The authors describe in detail what the important concept of ‘empowering a community’ entails and what this means for active engagement of all individuals as well as what this suggests for the involvement of nursing expertise. The authors analyze the new model for clinical decision making (the Community Assessment, Intervention, and Empowerment Model) and describe its importance. Readers will likely find some of the language difficult to follow and may be confused by the use of “will” as opposed to “does” throughout the manuscript. In sum, the manuscript includes adequate details, important topics, and relevant sources, and the plan is high level.

Author reply:

Response to Reviewer 2 Comments

Point 1: Readers will likely find some of the language difficult to follow and may be confused by the use of “will” as opposed to “does” throughout the manuscript.
Response 1: Thank you for your constructive feedback. We have greatly appreciated your comments and input throughout our manuscript as we think you antecipated the potential relevance of nursing expertise in the field.

Our efforts are focused on ensuring excellency in all levels of our academic work, therefore we have reviewed our manuscript accordingly, as suggested.

We have requested a revision from an English native lecturer to comply with your request, ensuring excellent language standards. However, we wish to remind that the use of “will” is supported by the fact that our manuscript only represents a protocol, which precedes the formal scoping review. The authors wish to demonstrate to the reader their intentions for the future scoping review.
We leave you with two examples of RCTs protocols where the logic is the same:
https://journals.lww.com/pedpt/Abstract/2022/01000/Effectiveness_of_Modified_Sports_for_Children_and.22.aspx
https://avr.tums.ac.ir/index.php/avr/article/view/966

Conclusions:
We feel that you have added great value to our manuscript and we hope we have responded accordingly to your valid points.

Reviewer 3 Report

The article describes a current topic (climate change, adaptation) and uses appropriate methods for this review report. The article makes a valuable contribution to this. The article is logically structured and written in an easily understandable way. The article describes the data collection in a clearly comprehensible way. The review contributes to mapping community actions in preventing climate change considering papers in English and Portuguese.

Minor remarks:

L. 74: many animals have died during the fire events. But do you have references that they are also extinct?

L. 75-77 not every extreme event is due to climate change, as many forest fires are caused by arson

L. 81-83 “Covid 19, for instance, the current pandemic, has also 80 taken advantage of a planet where climate change has not been properly acknowledged 81 [9].” Your reference “

Pereira, M., Calado, T., Da Camara, C., Calheiros, T. Effects of regional climate change on rural fires in Portugal. Climate Re-275 search. 2013; Vol 57: 187-200.“ doesn´t fit to Covid 19

L. 132: “What are the characteristics of these community actions to prevent/reduce climate change?” Please comment on what you are focusing on adaptation or mitigation

L. 183 “What are the characteristics of these community actions to prevent/reduce 132 climate change?” try to avoid repetition

Author reply:

Response to Reviewer 3 Comments

Dear Reviewer,
We would like to thank you for your kind comments and suggestions. In fact, we too feel very enthusiastic with the topic of choice, and we do appreciate your input in enhancing our paper.

Point 1: L. 74: many animals have died during the fire events. But do you have references that they are also extinct?
Response 1: Thank you for your note. It might be that the sentence is not clear enough to the reader, therefore we will add that, according to these authors, Connaker et al, it is clear that there was a record of forest fires in Australia, for instance, in 2019, causing destruction across the continent, killing billions of animals, which resulted in the extinction of some species as well – Please refer to new lines 73-75.

More than a year after, several other sources, such as Sciencenews for instance, (A year after Australia fires, hundreds of species may face extinction | Science News) reiterates that the toll on species is becoming increasingly clear – experts acknowledge that more than 500 species of plants and animals are now endangered or completely gone/extinct.
Another paper (https://pubmed.ncbi.nlm.nih.gov/34592040/Continental risk assessment for understudied taxa post-catastrophic wildfire indicates severe impacts on the Australian bee fauna - PubMed (nih.gov) refers to the risks that some of the 3 bilions vertebrates and 240 trilions invertebrates are at risk of extinction.

Point 2: L. 75-77 not every extreme event is due to climate change, as many forest fires are caused by arson
Response 2: Thank you for raising such a valid point. Although arson is a factor to have in consideration, July and August, the hottest months in the year in Portugal are solely accountable for 71% of all fire events, throughout the year. This idea was not further expanded due to the word limit that authors must comply with in the present protocol, however, gives the reader a general idea that increasing temperatures throughout summertime represent a trigger to forest fires in Portugal and in other Mediterranean countries, which in itself induces climate change as well, for instance by deteriorating the quality of the air. Please refer to new lines 76-80.

Point 3: L. 81-83 “Covid 19, for instance, the current pandemic, has also 80 taken advantage of a planet where climate change has not been properly acknowledged 81 [9].” Your reference “
Pereira, M., Calado, T., Da Camara, C., Calheiros, T. Effects of regional climate change on rural fires in Portugal. Climate Re-275 search. 2013; Vol 57: 187-200.“ doesn´t fit to Covid 19
Response 3 : Thank you for this note, unfortunatelly an editing error, has been acknowledged and has been amended as per below:
L. 78 refers to [9], now amended.
L. 82– now line 84 refers to [8], now amended.

Point 4: L. 132: “What are the characteristics of these community actions to prevent/reduce climate change?” Please comment on what you are focusing on adaptation or mitigation
Response 4 : The authors acknowledged that dividing mitigation and adaptation could be counterproductive and lead to a misinformed view that addressing climate change means pursuing either one or another. According to several authors1, nations, institutions and communities should follow both approaches, and policymakers need to prioritize integrated efforts from both angles.2 Following your suggestion, we may reshape the question in a way that reflects this in a more accurate way, such as:

“• What are the characteristics of these community actions to prevent and address climate change using both adaptation and mitigation approaches?”- Please see review question addressed as suggested on new lines 138-139.
1 Rieckmann M, Hoff H, Bokop K. Effective Community-Academic Partnerships on Climate Change Adaption and Mitigation: Results of a European Delphi Study. Sustain Clim Chang. 2021 Apr 1;14(2): p.1. – Please see new reference added [22] on new line 308.

Point 5: L. 183 “What are the characteristics of these community actions to prevent/reduce 132 climate change?” try to avoid repetition
Response 5 : We appreciate your note, although we are not sure whether you refer to the sentence on lines 183-185. On previous line 183 it said, "Following the search, all relevant studies will be identified from 2005 onwards, since Kyoto Protocol was implemented, enclosing the first legally binding greenhouse gas reduction targets to reduce the industrialized countries’ overall gas emissions." – we have now amended this on the new line 192, by changing “reduce” to “lower”.
Please do let us know if you had other purpose for your comment, as we are happy to address it accordingly.
Conclusions:
We feel that you have added value to our manuscript and we thank you for your valid points, now addressed.

Round 2

Reviewer 1 Report

Based on the authors' response letter and the tracked change document, I support publication of the manuscript.